# The Latest Advancement in Pancreatic Ductal Adenocarcinoma Therapy: A Review Article for the Latest Guidelines and Novel Therapies

**DOI:** 10.3390/biomedicines9040389

**Published:** 2021-04-06

**Authors:** Marwa Elsayed, Maen Abdelrahim

**Affiliations:** 1School of Medicine, University of Missouri Kansas City, 2301 Holmes, St. Kansas City, MO 64018, USA; mmeq4f@umkc.edu; 2Houston Methodist Cancer Center, Houston Methodist Hospital, 6445 Main Street, Outpatient Center, 24th Floor, Houston, TX 77030, USA; 3Cockrell Center of Advanced Therapeutics Phase I Program, Houston Methodist Research Institute, Houston, TX 77030, USA; 4Weill Cornell Medical College, Institute of Academic Medicine, Houston, TX 77030, USA

**Keywords:** pancreatic ductal adenocarcinoma, pancreatic cancer, pancreatic cancer treatment, pancreatic cancer novel therapy

## Abstract

Pancreatic ductal adenocarcinoma (PDAC) is the fourth leading cause of cancer deaths in the US, and it is expected to be the second leading cause of cancer deaths by 2030. The lack of effective early screening tests and alarming symptoms with early undetectable micro-metastasis at the time of presentation play a vital role in the high death rate from pancreatic cancer. In addition to this, the low mutation burden in pancreatic cancer, low immunological profile, dense tumorigenesis stroma, and decreased tumor sensitivity to cytotoxic drugs contribute to the low survival rates in PDAC patients. Despite breakthroughs in chemotherapeutic and immunotherapeutic drugs, pancreatic cancer remains one of the solid tumors that exhibit meager curative rates. Therefore, researchers must dedicate more effort to understanding the pathology and immunological behavior of PDAC, in addition to properly utilizing more advanced screening modalities and new therapeutic agents. In our review, we focus mainly on the latest updates from clinical guidelines and novel therapies that have been recently investigated or are under investigation for PDAC. We used PubMed as a search tool for finding original research articles addressing the latest developments in diagnosing and treating PDAC. Additionally, we also used the clinical trials published on clinicaltrialsgov as sources for our data.

## 1. Introduction

According to the CDC’s latest data, pancreatic ductal adenocarcinoma (PDAC) is the fourth leading cause of cancer deaths in the US [1], and it is expected to become the second leading cause of cancer death by 2030 [2]. In addition to the late presentation of the disease and lack of early alarming symptoms, a large majority of patients with pancreatic cancer present with either locally advanced or metastatic disease upon diagnosis; all of these factors contribute to the projecting burden of death by PDAC [3]. Furthermore, PDAC patients have an average 5-year survival rate of 10% [4]. Moreover, the deep location of the pancreas, aggressive nature of PDAC, and lack of reliable screening tests make it very hard for a primary physicians to catch and diagnose the disease at earlier and more manageable stages. PDAC commonly occurs as a sequence of accumulating genetic mutations in somatic cells, and less commonly, due to mutations in germline cells, particularly the case in familial PDAC [5,6]. With the advantage of next-generation gene sequencing, the PDAC pathophysiology is now better understood, and there is an emerging need for adding pathological classification to the current clinical and anatomical staging to formulate more personalized treatment plans [7,8]. Over the last decade, there has been several breakthroughs in early screening modalities and cancer therapeutics, particularly in the context of gastrointestinal malignancies; however, these have not matched the increment in both incidence and mortality rates of PDAC. Therefore, finding new strategies and therapeutic agents to improve the outcomes and survival of PDAC is an urgent need.

## 2. Treatment of PDAC

The only curative therapy for PDAC is surgical excision [9]; however, only 10–20% of patients are found to be eligible for resection at the time of diagnosis. More than 50% of patients are found to have metastatic disease at the first presentation [10]. To date, treatment approaches for patients with PDAC depend on the clinical and anatomical stage of the disease [11]. Notably, a great deal of ongoing studies are relying on the molecular pathology of PDAC to identify individualized, tailored plans for PDAC patients [12,13,14]. The initial workup process of PDAC aims to confirm the diagnosis and to look for evidence of distant metastasis [15,16]. Using imaging modalities, non-metastatic PDAC can be classified into resectable, borderline resectable, and locally advanced PDAC based on the involvement of the surrounding arterial (superior mesenteric artery, common hepatic artery, and celiac axis) and venous (superior mesenteric vein or portal vein) structures, and other nearby organs and lymph nodes [17,18]. According to the National Comprehensive Cancer Network (NCCN) guidelines published in 2017, certain criteria should be applied to obtain eligibility for surgical resection; the main goal is to involve as many patients as possible under the umbrella of surgical treatment [19]. The treatment outcome and the median overall survival depends in the first place on the stage of the PDAC, as mentioned in Figure 1.

### 2.1. Resectable PDAC

The criteria for resectability are summarized in Table 1 based on the latest recommendations from the NCCN, and AHPBA/SSO/SSAT Consensus (Americas HepatoPancreato-Biliary Association/Society of Surgical Oncology/Society for Surgery of the Alimentary Tract) [19,20]. Despite the remarkable development of surgical options and procedures for pancreatic cancer resection [9], a large number of patients will develop local recurrence or distant metastasis, which supports the theory of non-evident micro-metastases at the time of surgical resection [21,22]. This fact has prompted medical researchers to investigate the role of adjuvant chemotherapy as a mandatory course of treatment in patients with resectable PDAC. CONKO-001 was a large randomized multicenter clinical trial aimed at evaluating the efficacy and toxicity of gemcitabine in patients with pancreatic cancer after surgical resection for resectable PDAC [23,24]. A total of 354 patients were randomized in a 1:1 fashion to receive surgical treatment followed by gemcitabine for 6 months (treatment group) or surgical resection alone (observation group). The median overall survival (OS) was longer in the treatment group than in the observation group (22.8 months vs. 20.2 months; HR (Hazard Ratio), 0.76; *p*, 0.01), and the progression-free survival (PFS) was also favorable in the treatment group compared with the observation group (PFS, 13.4 months vs. 6.7 months; HR, 0.55; *p* < 0.001).

Fluoropyrimidines-based therapy, particularly with 5 Fluorouracil + Leucovorin (FU/LV), has been found to have a median OS that is comparable to gemcitabine when used as adjuvant therapy (based on the results of the multicenter randomized European Study Group for Pancreatic Cancer (ESPAC)-III trial) [25]. The reported median OS was 23 months for FU/LV (95% CI, 21.1–25.0) vs. 23.6 months for gemcitabine (95% CI, 21.4–26.4) with no meaningful clinical or statistical significance (HR, 0.94; 95% CI, 0.81–1.08; *p*, 0.39).

Interestingly, ESPAC-III proved the importance of completing the full adjuvant chemotherapy course rather than focusing on the early commencing of chemotherapy after surgical intervention. Subgroup analysis showed increased median OS in patients who completed all six planned cycles of treatment compared to patients who did not; 28.0 months (95% CI, 26.1–30.9) vs. 14.6 months (95% CI, 12.5–16.9), respectively; (HR, 0.516; 95% CI, 0.44–0.60; *p*, 0.001) [26]. On the other hand, initiation of chemotherapy within 8 weeks of surgery did not show any difference in OS compared to initiation of therapy after 8 weeks; median survival was 22.6 months (95% CI, 21.3–25.5) compared with 24.2 months (95% CI, 22.3–26.4), respectively (HR, 0.946; 95% CI, 0.82 to 1.09; *p*, 0.44).

Though completing the full course of adjuvant chemotherapy should be a target, it is important to keep in mind that this might not be achievable in all patients due to underlying comorbidities and their nutritional and functional status before and after the surgery.

Moreover, the phase III ESPAC-4 trial explored the efficacy of combined adjuvant chemotherapy treatment with gemcitabine as a base for the chemotherapy regimen [27]. A total of 730 patients, with a median follow-up of 43.2 months, were randomized to either gemcitabine in combination with capecitabine or gemcitabine alone groups. The median OS was 28 months in the combination group and 25.5 months in the gemcitabine alone group (HR, 0.82; 95% CI, 0.68–0.98; *p*, 0.032). The 5-year survival rate was significantly greater in the combination group than in the gemcitabine alone group (28.8% vs. 16.3%). These findings demonstrate that the combined use of capecitabine and gemcitabine is better than gemcitabine alone because of the synergistic effect between gemcitabine and capecitabine on the DNA thymidylate enzyme [28]. APACT is another phase III multicenter, open-label, randomized trial which investigated the role of gemcitabine-nab-paclitaxel (GnP) vs. gemcitabine alone as an adjuvant therapy for resectable PDAC. Though GnP was able to provide a longer median OS compared to gemcitabine alone (40.4 vs. 36.2 months, respectively; *p* = 0.045), this was not the case in terms of disease-free survival (DFS), where there was no clinical significance between GnP vs. gemcitabine alone (19.4 vs. 18.8 months, respectively; *p* = 0.1824) [29].

PRODIGE-24, a multicenter, randomized trial that was designed to evaluate the efficacy of modified folinic acid, fluorouracil, irinotecan, and oxaliplatin (FOLFIRINOX) (mFOLFIRINOX; FOLFIRINOX without the bolus dose of 5FU) compared to single-agent gemcitabine as an adjuvant treatment for resectable PDAC in patients with Eastern Cooperative Oncology Group (ECOG) performance status ≤ 1 [30]. A total of 493 patients were randomized into groups either receiving mFOLFIRINOX or gemcitabine with a median follow-up duration of 33.6 months. M.FOLFIRINOX was able to achieve a better median OS compared to gemcitabine alone; OS was 54.4 months vs. 35.0 months, respectively (HR, 0.64; 95% CI, 0.48–0.86; *p*, 0.003), and favorable PFS (21.6 months vs. and 12.8 months, respectively; HR, 0.58; 95% CI, 0.46–0.73; *p* < 0.001). It is essential to mention that mFOLFIRINOX was associated with higher rates of grade three to four adverse events compared to gemcitabine (75.9% vs. 52.9%, respectively); however, all adverse events were reversible, except for oxaliplatin related neurotoxicity, which persists in two patients in mFOLFIRINOX group.

To date, chemoradiotherapy has not been proven to play a major role in resectable PDAC. In 2004, ESPAC-1, a multicenter randomized trial with a 2 × 2 design, randomly assigned 145 patients to receive chemoradiotherapy (alone or with adjuvant chemotherapy) and 144 patients not to receive chemoradiotherapy (they received either chemotherapy alone or none), with a median follow-up of 47 months [31]. This study showed favorable outcomes for chemotherapy with a median OS of 21.6 months (95% CI, 16.5–22.7), while chemoradiotherapy was associated with worse outcomes with a median OS of 15.9 months (95% CI, 13.7–19.9).

The use of neoadjuvant therapy (NAT) in patients with the clear resectable disease is controversial and still under investigation. Theoretically, NAT could have favorable effects on free margin resection (R0), nodal staging, and non-visible microscopic metastasis [32]; however, studies have shown that postponing surgical resection for NAT might have a negative impact on the outcome and OS due to local disease progression, the deleterious effects of systemic chemotherapy, and adverse events of NAT, which could result in patients not being fit for surgical resection later on [33].

Ongoing studies are investigating the role of upfront NAT in patients with resectable disease. S1505 SWAG study, a phase II randomized clinical trial, is currently investigating the role of preoperative mFOLFIRINOX vs. GnP as an NAT in patients with resectable disease (clinicaltrials.gov (accessed on 4 April 2021) NCT02562716) [34]. Another ongoing trial, a phase III NEOPA study, recruited 410 patients to compare the 3-year survival rates between patients with resectable diseases who received preoperative chemoradiotherapy as an adjuvant treatment and those who underwent upfront surgery [35].

Both the National Comprehensive Cancer Network (NCCN) and the American Society of Clinical Oncology (ASCO) recommend upfront surgery followed by 6 months of adjuvant chemotherapy in the setting of resectable PDAC, but both recommend against NAT in the same setting, except for high-risk populations (for example, those with radiographic findings that lead to suspicion—but not diagnosis—of extra-pancreatic disease, i.e., markedly elevated CA19-9 levels, large primary tumors, large regional lymph nodes, excessive weight loss, or extreme pain) [19,36]. To date, there is no universal recommendation for one certain regimen of adjuvant chemotherapy over others. However, gemcitabine alone, gemcitabine in combination with capecitabine, continuous infusion 5-FU, or 5-FU/Leucovorin is the category one recommendation based on the latest NCCT guidelines [19].

### 2.2. Borderline Resectable/Locally Advanced PDAC

NAT plays an important role in borderline resectable PDAC. It not only leads to tumor shrinkage and makes tumors more amenable for surgical resection—with fewer complications and increased chance of achieving free margin resection (R0)—but also minimizes early non-detectable microscopic metastasis and decreases the load of lymph node involvement, thereby improving OS and outcomes [32,37,38,39].

The role of NAT has recently been investigated in multiple clinical trials, particularly after the new surgical consensus for PDAC treatment was released in 2009. A meta-analysis of 96 studies, including 5520 patients, investigated the role of NAT in resectable, borderline resectable, and locally advanced diseases [40]. The results showed favorable outcomes for NAT, particularly in borderline resectable and locally advanced diseases, with resection rates and R0 resection rates of 70% and 84% for borderline resectable disease and 32% and 82% for locally advanced disease, respectively. Although these results are encouraging because achieving R0 resection has been considered an independent prognostic factor for OS and disease recurrence [41,42], it is important to note the heterogeneity of the included studies and lack of standardized individual data in this meta-analysis.

The suitability of the NAT chemotherapeutic regimen is a current research topic, and most studies suggest the use of either FOLFIRIONX or GnP as the first line of treatment for locally advanced PDAC. For example, a systematic meta-analysis demonstrated better median OS and PFS in patients who received FOLFIRINOX (median OS, 24.2 months (95% CI, 21.6–26.8); median PFS, 15.0 months (95% CI, 13.8–16.2)) [43]

It is important to keep in mind, that up to now, there has been no head-to-head study comparing FOLFIRINOX to GnP as an NAT in the setting of borderline resectable PDAC or locally advanced PDAC.

The preliminary findings from the PREOPNAC study are positive for the role of preoperative chemoradiotherapy (CRT) in borderline PDAC [44]. A total of 246 patients were randomized into groups receiving either preoperative gemcitabine with radiotherapy or upfront surgery, and both groups received adjuvant gemcitabine following surgery. Although CRT improved the R0 resection rates, PFS rates, lymph node involvement, perineural invasion, and venous invasion, there was no statistically significant difference in the median OS between the two groups (median OS, 16.0 months vs. 14.3 months; HR, 0.78; 95% CI, 0.58–1.05; *p*, 0.096).

The recently released ESPAC-5F study was a four-arm randomized trial that aimed to compare the roles of FOLOFIRIONX, gemcitabine-capecitabine, CRT, and upfront surgery in borderline resectable settings of PDAC. All groups received adjuvant chemotherapy after surgical resection. Although there was no difference in the first primary endpoint (resection rate) between the NAT group and the upfront surgery group, the second endpoint was favorable in the NAT group compared with the upfront surgery group (12-month survival rate: 77% (95% CI, 27–64%) vs. 42% (95% CI, 66–89%)) [45]. Among the four groups, the FOLFIRINOX group showed a better 12-month survival rate with a small increased risk of manageable toxicities (84% (95% CI, 70–100)), followed by the gemcitabine-capecitabine group (79% (95% CI 63–100)) with a small increased risk of manageable toxicities, with the CRT group ranking last (64% (95% CI, 43–95%)).

The latest guidelines from the NCCN, which are similar to the recent 2019 recommendations from the ASCO, recommend no upfront surgery for borderline resectable and locally advanced diseases, and state that FOLFIRINOX, gemcitabine-nab-paclitaxel, and gemcitabine-cisplatin (particularly in patients with DNA repair mutation) are acceptable regimens [19,36]. CRT, as an NAT, is preserved for patients who present with poorly uncontrolled pain, local invasions with bleeding, deteriorating performance status, and local disease progression despite chemotherapy, without evidence of distant metastasis [19,36].

### 2.3. Metastatic PDAC

Metastatic PDAC has the worst prognosis among the various stages of PDAC, with a median 1-year survival rate of 7% [46]. The treatment choices for metastatic PDAC are limited (Table 2), and the main aim of therapy is palliation of symptoms, particularly in patients with poor performance status at the time of diagnosis. Since 1997, gemcitabine has been considered the first line of treatment for metastatic PDAC, based on findings from published studies which proved the superiority of gemcitabine over 5FU in terms of improving OS in patients with metastatic PDAC [47,48].

The notion of increasing the efficacy and outcomes of gemcitabine by adding another cytotoxic drug has been investigated over the last two decades. Gemcitabine in combination with other fluoropyrimidines-based anti-tumor drugs has been demonstrated to have more favorable outcomes than gemcitabine alone [49]. For example, Cunningham et al. showed that combined Gem–Cap had better median OS and PFS than Gem alone in the setting of advanced PDAC [50]. The median OS was 7.1 months for GEM–CAP and 6.2 months for GEM (HR, 0.86; 95% CI, 0.72–1.02; *p* < 0.08). The 1-year OS rates were 24.3% for GEM-CAP and 22% for GEM. PFS was 5.3 months in the GEM–CAP group and 3.8 months in the GEM group (HR, 0.78; 95% CI, 0.66–0.93; *p* < 0.004). The 12-month PFS rates were 13.9% for GEM–CAP and 8.4% for GEM.

In the phase III National Cancer Institute of Canada – Clinical Trials Group (NCIC CTG) PA.3 study, the combination of erlotinib with gemcitabine was associated with a nonmeaningful clinical improvement in the median OS (OS: 6.24 months in the combined gemcitabine and erlotinib group vs. 5.91 months in the gemcitabine group; HR, 0.82; 95% CI, 0.69–0.99; *p*, 0.04) [51,52]. These parameters were slightly better in patients with higher functional status and lower pain scores at the time of diagnosis. These findings indicate that gemcitabine, in combination with other cytotoxic drugs, shows only minimal improvement compared to gemcitabine alone. However, combination of albumin-bound paclitaxel with gemcitabine was a breakthrough in the gemcitabine-based therapy for advanced PDAC. The phase III Molecular Profiling-based Assignment of Cancer Therapy (MPACT) trial demonstrated that a combination of gemcitabine with nab-paclitaxel had a better OS than gemcitabine alone in patients with metastatic PDAC (median OS, 8.7 months vs. 6.6 months; HR for death, 0.72; 95% CI, 0.62–0.83; *p* < 0.001) [53]. Gem-nab-paclitaxel treatment improved OS even in patients with the high-risk features such as elevated levels of CA19-9 (HR, 0.612, 95% CI, 0.49–0.76, *p* < 0.001).

Despite the previous results from the phase III MPACT trial, FOLFIRINOX is still considered to be superior to Gemcitabine based regimens for patients with metastatic PDAC with a good ECOG performance scale (PS) because of the more favorable median OS and more tolerability of adverse events in these population. In 2011, phase III PRODIGE, a randomized study, recruited 342 patients with advanced PDAC who had an ECOG performance scale of ≤1, and they were randomized to either FOLFIRINOX or Gemcitabine treatment groups for 6 months. The FOLFIRINOX group showed median OS of 11.8 months and PFS of 6.6 months, while the gemcitabine group showed median OS of 6.8 months and PFS of 3.3 months (HR, 0.57; 95% CI, 0.45–0.73; *p* < 0.001) [54]. Although the FOLFIRINOX group was associated with more toxicities, particularly febrile neutropenia, the 6-month degradation of life was better in the FOLFIRINOX group than in the gemcitabine alone group (31% vs. 66%; HR, 0.47; 95% CI, 0.30–0.70; *p* < 0.001). The higher rates of median OS and PFS that were recorded in the FOLOFIRINOX group might be due to the use of Irinotecan, which has activity against PDAC by itself, and synergistic activity when given prior to fluorouracil [55,56]. In addition, platinum-based oxaliplatin is also more effective in the presence of fluorouracil [57].

To date, no head-to-head randomized controlled trials have compared FOLFIRINOX and GnP in terms of OS and PFS. A retrospective cohort study, which randomly assigned 216 patients to either the FOLFIRINOX (*n* = 109) or GnP (*n* = 107) group, showed a favorable outcome in the FOLFIRINOX group compared to the GnP group (median OS, 14 months (95% CI, 10–21) vs. 9 months (95% CI, 8–12); *p*, 0.008), even after adjusting for age, number of metastatic sites, liver metastases, peritoneal carcinomatosis, and CA19.9 level at baseline (HR, 0.67; *p*, 0.097) [58]. However, this clinically significant result is not only attributed to the FOLFIRINOX therapy, since more patients in this group received the second line therapy of GnP (72.0% vs. 57.8%, respectively; *p* = 0.042), indicating that this higher OS in the FOLFIRINOX group was partially due to the use of GpN as a second line of treatment. The sequence FOLFIRINOX (FFX) followed by GnP (FFX–GnP) was feasible in a higher proportion (43.0%) than the reverse sequence (GnP–FFX) (12.8%; *p* < 0.001). In addition, adding GnP as a sequence therapy after FOLFIRINOX failure was found to improve the median OS (7.6 months) and median PFS (3.8 months). The total median OS was 14.2 months (95% CI, 10.6–15.1), and the total median PFS was 9.3 months (95% CI, 7.5–12.4) from the first dose of FILFIRINOX. This improvement in the median OS was associated with increased grade three and four adverse events, particularly hematological side effects and peripheral sensory neuropathies.

Another large Korean retrospective study compared FOLFIRINOX and GnP as a first-line treatment for metastatic PDAC [59]; the findings favored GnP over FOLFIRINOX, with a median OS of 12.1 (95% CI, 10.7–undetermined) and 10.7 months (95% CI, 9.1–12.3), PFS of 8.0 and 8.4 months (*p* = 0.134), and objective response rates of 33.7% and 46.9% (*p* = 0.067) in the GnP and FOLFIRINOX groups, respectively; however, these results were not statistically significant. Moreover, modified FOLFIRINOX (FOLFIRINOX without 5FU bolus) and GpN therapies have been found to have similar outcomes in terms of median OS and PFS, with a very similar profile of adverse events

Based on the results of Watanabe. K. et al., GpN therapy provided better median OS, PFS, and 12-month survival rates compared to mFOLFIRINOX. The median OS was 14.0 months (95% CI, 12.2—not reached) vs. 11.5 months (95% CI, 9.7–16.8), respectively, the PFS was 6.5 months (95% CI, 6.1–7.9) vs. 5.7 months (95% CI, 3.4–7.1), respectively, and the 12-month survival rate was 44% vs. 67%, respectively (*p*, 0.0006) [60]. A recent meta-analysis which investigated twenty-two retrospective studies that recruited 6351 patients showed similar outcomes in the median OS and PFS between GnP and mFOLFIRINOX, while also demonstrating similar toxicity profiles [61].

Based on the above results, both FOLFIRINOX and GnP are recommended for patients with ECOG PS of zero or one as the first line therapy for metastatic PDAC, and gemcitabine alone is recommended as the first line therapy for patients with an ECOG PS of two or above [19,36]. Based on encouraging results from clinical trials, sequencing between fluoropyrimidine based-therapy and gemcitabine based-therapy is currently recommended as a second line therapy for progressive metastatic PDAC (based on the choice of the first line therapy and the level of clinical response and performance status) by both the NCCN and ASCO guidelines [62,63,64].

New combinations of cytotoxic drugs as a second line therapy for progressive metastatic PDAC have been investigated for a while. NAPOLI-1 was a randomized, multicenter, open-label trial that was designed to investigate the role of liposomal irinotecan (nal-IRI) + 5-flouruoracil (5FU) + leucovorin (LV) vs. 5FU + LV vs. liposomal irinotecan alone as a second line therapy in gemcitabine-resistant metastatic PDAC. The nabIRI + 5FU + LV combination showed better OS than 5FU + LV (OS, 6.2 vs. 4.2 months; HR, 0.63; 95% CI, 0.47–0.85; *p*, 0.002). There was no superiority of nal-IRI over 5FU + LV. The median PFS was 3.1 months in patients receiving nal-IRI + 5-FU + LV and 1.5 months in those receiving 5-FU + LV (HR, 0.57; 95% CI, 0.43–0.76; *p* < 0.0001); however, younger age, better performance status, absence of liver metastasis with lower CA19-9 levels, and a lower neutrophil to lymphocyte ratio (<5) contributed to the better outcome in the nab-IRI + 5FU + LV group [65]. Currently, nal-IRI in combination with Oxaliplatin and 5FU/LV is being investigated as a first line therapy for newly diagnosed metastatic PDAC vs. GnP in the NAPOLI 3 trial; an open-label randomized multicenter phase III clinical trial (NCT04083235).

## 3. Novel Emerging Therapies

A better understanding of the molecular pathology of PDAC could allow researchers to investigate more new modalities for PDAC treatment [66,67,68]. Most of these strategies are still under investigation, and some of them have been aborted because of negative results. As a general rule, the combination of these new modalities (one targeted molecule agent and one cytotoxic agent) leads to better outcomes [69]. Figure 2 summarizes the currently available novel therapies for PDAC.

### 3.1. Targeting KRAS Pathway

KRAS (Kirsten-Ras protein) is the most common somatic mutation in PDAC and can be found in >90% of patients [70]. KRAS, an oncogenic gene, is crucial for not only adenoma initiation but also maintenance of growth and architecture of tumor through other signaling pathways [71]. Mutated KRAS was detected in >90% of early low grade pancreatic intra-epithelial lesions (PanIN) [72].

The KRAS mutation in PDAC is mostly due to a point mutation which mainly occurs at codons 12, 13, and 61. The point mutation of G12D (glutamine (G) substituted for aspartic acid (D)) is more frequently seen than other amino acid substitutions, e.g., G12C (glutamine (G) substituted by cysteine (C)) [72,73]. Despite this advancement in KRAS mutation molecular analysis, investigators are struggling to invent an effective stable molecule that can directly target mutated KRAS.

The results from preclinical studies evaluating drugs that target mutated KRAS{G12C} are encouraging; ARS-1620 is a novel inhibitor of KRAS{G12C} that showed positive results as an antitumor agent, with a safe profile [74]. AMG-510 is another novel therapy that targets KRAS{G12C} and has been studied in clinical trials. This has revealed promising results not only related to its function as an antitumor drug, but it has also shown an immune modulator role, particularly when given in combination with Immune Check point Inhibitors (ICI) in mice modules [75,76,77]. Despite this progress in developing therapeutic agents that target the KRAS{G12C} mutation, it is worth mentioning that this mutation only occurs at a meager rate in PDAC (1–2%) [78,79].

Another new strategy is to develop specific siRNAs that target KRAS{g12d}, the most common somatic point mutation among PDAC associated with mutated KRAS. This new therapy is under investigation as a single agent in a phase I trial (NCT03608631) and in combination with GnP in a phase II trial (NCT01676259) [80,81].

Since directly targeting KRAS is not yet clinically feasible, studies have started targeting either upstream or downstream molecules that are involved in KRAS pathway activation. Larotrectinib and entrectinib have been approved by the FDA for solid tumors that harbor Neurotrophic Tyrosine Kinase (NTRK) [82,83]. The NTRK fusion mutation is responsible for producing the TRK fusion protein, a molecule that activates downstream signals that are responsible for both the Mitogen activated Protein Kinase- Extracellular Regulated Kinase (MEK-ERK) and Phosphoinositide3 Kinase-Serine Threonine (PI3K-AKT) signaling pathways. These pathways play a vital role in tumor growth maintenance [84]. The NTRK fusion mutation is only found in <1% of PDCA patients [85]. The downside of using these novel therapies is increased resistance and disease progression due to acquired mutations in NTRK during treatment; thus, researchers are investigating the role of second generation NTRK targeting therapies to overcome these limitations [86,87]. Preclinical trial results from such studies are encouraging.

Additionally, the PI3K-AKT-mTOR pathway has been an area of interest in the modern era of research. PI3K-AKT-mTOR is the downstream signal for KRAS pathway activation; the unopposed activation of the PI3K-AKT-mTOR pathway can have a tremendous effect on cell proliferation, growth, and enhancing cellular adaptive mechanisms [88]. Moreover, PI3K was recently suggested to be capable of boosting the immune resistance properties of PDAC [89,90]. Although PI3K inhibitors have been approved by the Food and Drug Administration (FDA) for some solid cancers, such as breast cancer [91], this is not the case for PDAC. PI3K inhibitors have failed to demonstrate any effective results in treating pancreatic cancer due to increased numbers of drug-related adverse events during therapy [92,93]. Recently, researchers investigated the effect of adding another pathway inhibitor—e.g., a MAPK inhibitor to the PI3K inhibitor—while treating solid cancers, which has greatly improved the efficacy; however, there was an increase in dose-limiting toxicity due to a higher incidence of grade ≥ 3 drug-related toxicity [94]. It is important to mention that the full mechanism of the PI3K-AKT-mTOR pathway in initiating and maintaining PDAC is not yet fully understood, and a great deal of ongoing clinical trials are currently investigating the role of the PI3K inhibitor in treating PDAC (e.g., NCT02981342, NCT03065062, NCT02155088). On the other hand, patient selection and stratification based on genetic profiles before initiating therapy with PI3K is another area to be explored. Theoretically, administering PI3K inhibitors to patients who have a mutated PI3K pathway will yield the maximum benefit of this therapy, and this has been demonstrated in multiple clinical trials; however, cases have been reported where patients who lack this mutation can still show response to PI3K inhibitors and vice versa [95].

As with the PI3K-AKT-mTOR pathway, the Rapidly Accelerated Fibrosarcoma- Mitogen activated Protein Kinase- Extracellular Regulated Kinase (RAF-MEK-ERK) is another down-stream pathway of the KRAS activation system which has been found to play a major role in PDAC initiation and maintenance [96]. Clinical studies investigating the role of BRAF and MEK inhibitors, either as a monotherapy or in combination with other PDAC chemotherapeutic agents, failed to show positive outcomes [97,98,99,100]. It worth mentioning that using a combination of BRAF and MEK inhibitors could have a synergistic effect when treating PDAC [101], however, this area requires further research via clinical trials.

Interestingly, NRG1 fusion is more active in patients with KRAS wild type, and its oncogenic properties are due to downstream activation of Epidermal Growth Factor Receptor (EGFR) via the production of neuregulin-1, a ligand for both erb-EGFR3 and erb-EGFR4 receptor tyrosine kinases [102,103]. EGFR inhibitors have been found to show better outcomes in patients with KRAS wild type. ERLOTINIB (anti-EGFR monoclonal therapy) plus gemcitabine has been approved by the FDA as a first line therapy for metastatic PDAC, after showing a minor improvement in OS compared to gemcitabine alone [52]. Nimotuzumab (another monoclonal antibody against EGFR) in combination with gemcitabine showed a significant improvement in OS compared to gemcitabine alone for metastatic PDAC, particularly in patients with the wild type KRAS (median OS: 11.6 months vs. 5.6 months) [104].

### 3.2. Targeting DNA Damage Repair (DDR) Pathway

Mutations in DNA Damage Repair (DDR) genes are present in up to 24% of PDAC [105], with BRCA1/2, PALB2, ATM, MLH1, MSH2, and MSH6 being the most commonly reported ones [106]. PDAC associated with mutations in the DDR gene is more responsive to platinum-based therapy in combination with other cytotoxic chemotherapies [107] and poly ADP-ribose polymerase (PARP) inhibitors [108]. NCCN recommends treating patients who have inherited mutations in the DDR pathway with platinum-based chemotherapy [19]; this therapy has been proven to improve the median OS in PDAC cases, especially at advanced stages (patients with platinum-based therapy vs. those without 22 months (95% CI, 6–27) vs. 9 months (95% CI, 4–12)) [109]. A similar result was detected in patients with metastatic PDAC and a positive family history of pancreatic, breast, or ovarian cancers in a retrospective study from Johns Hopkins University School of Medicine [110]. In addition, Olaparib, a PARP inhibitor, was recently approved by the FDA as maintenance therapy for metastatic PDAC with a positive BRCA1/2 mutation. The POLO study is a pivotal study because it was the first to prove the benefit of targeted therapy against mutated genes in the subgroup population of PDAC. A total of 3135 patients with metastatic PDAC were screened for BRCA1/2 mutations, and 154 were randomized in a 3:2 fashion to either the oral Olaparib or placebo group as maintenance therapy after the last dose of platinum-based chemotherapy without disease progression. The primary endpoint was PFS. Patients who received Olaparib were able to achieve a PFS of 7.4 months compared to 3.8 months in the placebo group (complete/partial disease HR, 0.62; stable disease HR, 0.50) [111].

### 3.3. Targeting the Immune System

#### 3.3.1. Immune Check Point Inhibitors (ICI)

ICI have been found to be effective in solid tumors with a high tumor mutational burden (TMB) [112] that exhibit 3ry lymphoid structures, such as melanoma [113,114]. The findings of early trials using ICI for advanced pancreatic cancer were disappointing. Part of this owes to the paucity of the TMB in PDAC; the other part owes to using ICI as a monotherapy [115]. Although MMR deficiency (dMMR) is very rare in PDAC (~1%) [116], it is associated with a high TMB in PDAC. Pembrolizumab was approved by the FDA for all solid tumors, including PDAC, in patients with dMMR mutation [117,118]. Notably, PDCA has an immunosuppressive tumor microenvironment with limited numbers of cytotoxic T cells, which explains the lack of effect of ICI in PDAC as a monotherapy [78,79]. One of the strategies to enhance the mechanism of ICI in tumors with low immunogenicities, such as PDAC, is to combine them with other immunomodulatory agents. Phase I/II clinical trials aimed at investigating the effect of ICI in combination with other immunomodulatory agents (e.g., therapeutic vaccines, agnostic immunotherapy, and myeloid immunotherapy) with or without chemotherapy are summarized in Table 3.

#### 3.3.2. Vaccines

Two vaccines have been manufactured as therapeutic agents for advanced PDAC; GVAX, an irradiated tumor cell, is genetically treated to express granulocyte colony-stimulating factors to enhance profound T-cell production against a wide variety of tumor-specific antigens [119]; CRS-207, a mesothelin expressing live attenuated Listeria monocytogens, boosts the effect of GVAX [120]. Mesothelin is secreted abundantly in PDAC and plays a crucial role in cancer cell proliferation, distant metastasis facilitation, and tumor resistance to certain chemotherapeutics agents [121,122]. However, initial trials using a combination of these two vaccines failed to demonstrate an improvement in OS [123]; using these vaccines to boost the effect of ICI with chemotherapy is currently under investigation in phase II trials (NCT03006302 and NCT02243371).

#### 3.3.3. Chimeric Antigen Receptor T Cells (CAR-T Cells) Transfusion

CAR-T cell transfusion therapy has been demonstrated to have a high efficacy for treating hematological malignancies, and it has recently been approved by FDA for certain types of lymphoma and leukemia [124,125]. However, CAR-T cell engineering remains challenging for solid tumors, partly owing to the dense stroma surrounding these tumors (the opposite is true for hematological malignancies) and the heterogeneity of solid tumor cell epitopes, which make it difficult for CAR-T cells to target only the tumor cells without affecting the surrounding normal cells [126]. Meso-CAR T cells are CAR-T cells that target mesothelin in solid tumors [127]. Phase I trial investigating the safety and efficacy of meso-CAR T cells revealed satisfactory results, with a reported 69% decrease in tumor volume in one patient [128].

CD40, a member of the tumor necrosis factor receptor superfamily, is found on the surface of multiple immune cells, particularly B cells and antigen representing cells (APC) [129]. One strategy is to activate the CD40–CD40L interaction in APCs and enhance the effect of cytotoxic T cells against tumor antigens to enhance the host immune response against tumor cells; indeed, the CD40 agonist has been found to have independent T-cell anti-tumor effects by depleting the tumor stroma [130,131]. A phase Ib cohort study showed a response rate of 58% in 24 patients with untreated metastatic PDAC who received APX005M, an agonistic anti-CD40 antibody with Gem-nab-Pac, both with or without nivolumab (PD-1 monoclonal antibody). Additionally, a noticeable depletion of T cells and increasing macrophage infiltration into the PDAC stroma were recorded [132].

### 3.4. Targeting Tumor Metabolism

Preventing tumor cells from obtaining the required energy for survival and growth is under investigation. Targeting rate-limiting enzymes that are involved in the metabolism of tumor cells, in combination with cytotoxic drugs, has been found to be an effective strategy in preclinical trials. Devimistat, an antimitochondrial drug that prevents energy production from the tricarboxylic acid cycle by directly inhibiting pyruvate dehydrogenase and α-ketoglutarate dehydrogenase enzymes, was found to have a 61% response rate in PDAC in a small single-center trial when used in combination with mFOLFIRINOX [133]. Currently, Devimistat is under investigation with m.FOLFIRINOX in a phase III trial (NCT03504423) and with gemcitabine and nab-paclitaxel in a phase I trial (NCT03435289).

### 3.5. Targeting Tumor Stroma and Extracellular Matrix

The PDAC stroma is unique and plays a crucial role in supporting tumor growth, thereby promoting distant metastasis and increasing drug resistance [78]. The role of the PDAC stroma is driven mainly by the dense desmoplastic reaction (DR), which is composed of a cellular part (such as pancreatic stellate cells, tumor-associated macrophages, and mast cells) and an acellular part (such as hyaluronic acid, collagen I, and collagen III) [134,135]. Initial studies have demonstrated that targeting certain PDAC stromal components in combination with cytotoxic drugs could be an approach to yield better outcomes.

#### 3.5.1. Targeting Hyaluronic Acid

Increased hyaluronic acid (HA) in PDAC is associated with increased interstitial pressure, which decreases tumor perfusion and drug delivery [136]. A novel recombinant pegylated human hyaluronidase enzyme (PEGPH20), in combination with gemcitabine and nab-paclitaxel, was investigated for metastatic PDAC compared with gemcitabine and nab-paclitaxel alone [137]. Although the overall results from HALO-202—a phase II randomized trial—did not show a meaningful clinical difference in PFS between both groups, a subgroup analysis showed a statistical difference between both groups for patients with detected high levels of HA expression, with a PFS of 9.2 months in patients with high levels of HA and 5.2 months in those with low levels of HA (HR, 0.51; *p*, 0.049). HALO-301—a randomized phase III trial—was designed to examine the efficacy of PEGPH20 in combination with GnP in selected patients with advanced PDAC in the setting of high HA levels [138]; preliminary results have not yet been published. Disappointing results have been noticed when examining FOLFIRINOX in combination with PEGPH20 vs. FOLFIRINOX alone (median OS, 7.7 months vs. 14.4 months, respectively) in unselected patients with PDAC (unselected regarding their HA status). This study was terminated for accrual results and increased side effects in the PEGPH20 group. [139].

#### 3.5.2. Targeting Fibroblast Component

As mentioned previously, PDAC is characterized by a dense DR, which has been shown to play a vital role as a physical barrier that prevents drug delivery in preclinical models [140]. By adding targeted therapy against the hedgehog signaling pathway (SSH) to the conventional chemotherapeutic agents, the researchers hypothesized that this could significantly improve drug delivery and efficacy, leading to better treatment outcomes; however, the results from preclinical trials were disappointing. Researchers discovered that adding therapeutic agents that target SSH or stromal fibroblasts caused a depletion of stromal collagen content and promoted undifferentiated growth, which led to more aggressive behavior and reduced responsiveness by the tumor to the traditional therapeutics. Furthermore, PDAC with lower SSH expression has been found to have a more accelerated growth rate and fatal outcome. [141,142]. Clinical trials that have investigated the role of vismodegib—an SHH pathway inhibitor—in combination with GnP for untreated metastatic PDAC were terminated early due to similar negative results [143].

### 3.6. Targeting Immune Cells/Signals in PDAC Stroma

Myeloid-derived suppressor cells (MDS) and Treg cells both play important roles in facilitating the anti-tumor phenotype for PDAC by suppressing host immune responses against tumor cells, including decreasing anti-tumor cytotoxic CD8 T cell recruitment and upregulating PD1-PDL1 expression in tumor cells, which is a crucial step in facilitating tumor growth, distant metastasis, and increased drug resistance [144,145].

Drugs that target pro-MDS proteins, which facilitate the differentiation of myeloid precursors into MDS and Treg recruitments (such as focal adhesion kinase (FAK), Bruton tyrosine kinase (BTK), and CXCR4) [146,147,148,149,150,151,152,153,154,155], are currently under investigation with some promising preliminary results.

FAK inhibitors in combination with GnP showed positive results in tumor growth inhibition and tumor regrowth delay in in vivo and in vitro models [149,150]. The latest trials investigating the role of FAK inhibitors are listed in Table 4.

BTK plays an important role in activating CD20 B cells to differentiate into M2 phase MDS cells, which promote the protumor and anti-immune environment. Ibrutinib, a BTK inhibitor, blocks CD20 B cell differentiation into MDS cells. Preclinical trials showed that Ibrutinib promoted more differentiation towards anti-tumor cytotoxic CD8 T cells [151]; however, the results from the RESOLVE trial, which was designed to investigate the role of Ibrutinib in combination with G-nab-P, were disappointing [152]. Another phase Ib/II trial investigated the role of Ibrutinib in combination with Durvalumab (PD1L monoclonal antibody) and showed similar results, with limited effects on tumor growth [153].

CXCR4 interaction with CXCL12 has been found to promote MDS cell differentiation, which, in turn, promotes a more immune-suppressive environment in the PDAC stroma. Drugs that inhibit this interaction in combination with ICI were found to increase tumor infiltration of cytotoxic CD8 T cells and downregulation of MDS cells and Treg cells [154,155]. BL-8040 (motixafortide; a CXCR4 inhibitor) in combination with pembrolizumab was found to achieve the objective response rate, disease control rate, and median duration of response of 32%, 77%, and 7.8 months, respectively [155].

## 4. Conclusions

In summary, surgical treatment remains the only available curative therapy for PDAC. Adjuvant and neoadjuvant therapies represent the cornerstone for PDAC treatment, while a large number of novel targeted/immunomodulatory therapies—along with cytotoxic drugs—are lightening the road, with breakthroughs in PDAC management expected over the coming decade. In the era of precision medicine, and with advancements in next-generation gene sequencing, examining genetic mutations in PDAC is considered a mandatory step to formulate a well-tailored and personalized therapy plan. However, focusing on genetic mutations in PDAC alone is not sufficient; the role of epigenetics in PDAC is another area that requires further exploration through clinical trials. For the sake of depth and focus, the rle of epigenetics has not been discussed in this review article.

## Figures and Tables

**Figure 1 biomedicines-09-00389-f001:**
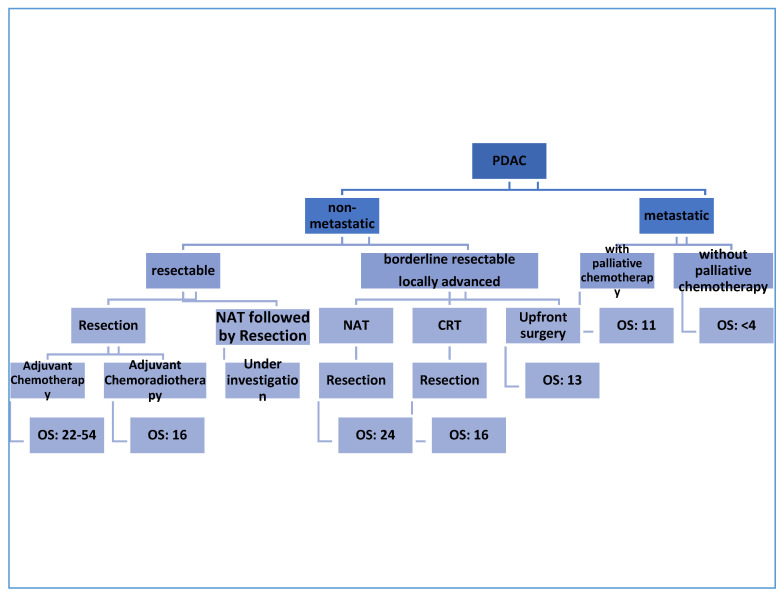
The outcomes of different treatment strategies in PDAC. OS: overall survival (measured in months); NAT: neoadjuvant chemotherapy.

**Figure 2 biomedicines-09-00389-f002:**
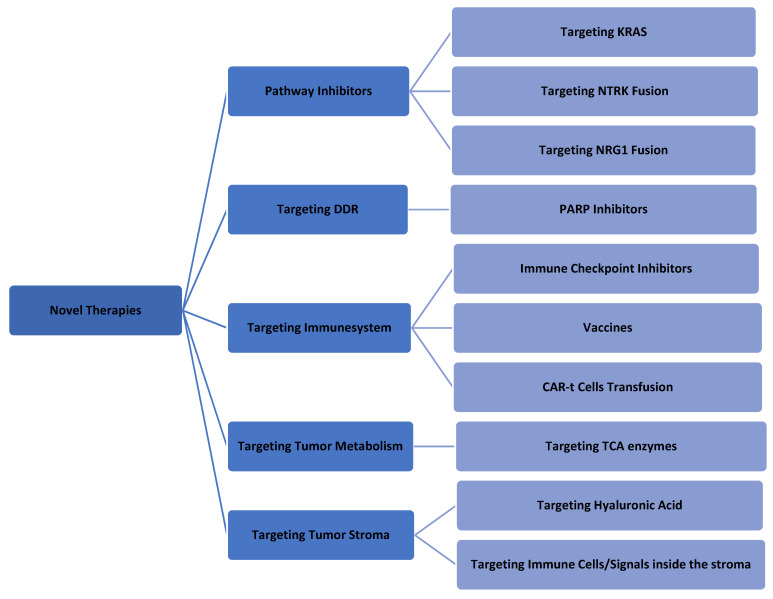
Summary of the last novel therapeutics in PDAC treatment.

**Table 1 biomedicines-09-00389-t001:** Resectability criteria based on NCCN (National Comprehensive Cancer Network) and AHPBA/SSO/SSAT (Americas HepatoPancreato-Biliary Association/Society of Surgical Oncology/Society for Surgery of the Alimentary Tract) Consensus.

Stage	SMA	CA	SMV
Resectable	No involvement	No involvement	No involvement or <180° contact without contour irregularities
Borderline Resectable	≤180° contact	Contact with CHA without extension to CA	>180° contact or ≤180° contact with contour irregularity, with short segment involvement and suitable proximal and distal vessel for reconstruction
		≤180° contact	
		≥180° contact without involvement of aorta	
Locally advanced	Distant metastasis	Distant metastasis	Distant metastasis
	>180° contact	>180° contact	Long segment involvement with difficult reconstruction
	Contact with 1st jejunal branch	Involvement of aorta	Contact with 1st draining jejunal branch

SMA: superior mesenteric artery; CA: celiac axis; CHA: common hepatic artery; SMV: superior mesenteric vein; PV: portal vein.

**Table 2 biomedicines-09-00389-t002:** The most common chemotherapy used in the treatment of metastatic pancreatic ductal adenocarcinoma (PDAC).

Chemotherapy	OS	PFS	Comments
Gemcitabine+	14 m	6.5 m	Best for ECOG 0-2
Nab-Paclitaxel			Category 1 recommendation
FOLFIRINOX	11.8 m	6.6 m	Best for ECOG 0-1
			Category 1A recommendation
Gemcitabine+	8.5 m	5.3 m	Best for ECOG 0-1
Capecitabine			Category 2A recommendation
Gemcitabine+	6.24 m	3.5 m	Best for ECOG 0-2
Erlotinib			Category 1 recommendation
Gemcitabine+	12 m		Recommended for patients with DDR inherited mutation
Cisplatin			
Gemcitabine	5.9 m	3.3 m	Best for ECOG 0-4
			Category 1 recommendation

ECOG: Eastern Cooperative Oncology Group Performance Status; DDR: DNA damage repair.

**Table 3 biomedicines-09-00389-t003:** Recent clinical trials designed to investigate the role of anti PD1/PDL1 monoclonal antibodies in association with other immunomodulatory therapies in PDAC treatment.

Drug Name	Immunomodulator Combination	Mechanism of Action	NCT No.	Phase	Additional Agents
**Anti-PD1**					
Pembrolizumab	BL-8040	CXCR4 antagonist	NCT02907099	II	
	BL-8040	CXCR4 antagonist	NCT02826486	II	
	Olaptesed Pegol	CXCL12 antagonist	NCT03168139	I/II	
	Epacadostat	IDO1 inhibitor	NCT03006302	II	CRS-207, CY/GVAX
	Young TIL	Tumor Infiltrating Lymphocytes	NCT01174121	II	
Nivolumab	Ipilimumab	CLTA-4 blockers	NCT01928394	I/II	
	Ipilimumab	CLTA-4 blockers	NCT02866383	II	RT
	Lirilumab	KIR2DL1/2/3 inhibitors	NCT01714739	I/II	Ipilimumab
	Cabiralizumab	CSF1R inhibitor	NCT03336216	II	Chemotherapy
	APX005M	CD40 agonist	NCT03214250	Ib/II	GnP
	CY/GVAX	Recombinant Vaccine	NCT02243371	II	CRS-207
**Anti-PDL1**					
Durvalumab	Pexidartinib	CSF1R inhibitor	NCT02777710	I	
	Galunisertib	TGFβ inhibitor	NCT02734160	I	
	Oleclumab	5′-Nucleotidase inhibitor	NCT03611556	I/II	Chemotherapy
Avelumab	Binimetinib	MEK inhibitor	NCT03637491	II	Talazoparib

PD1: Programmed cell death protein 1; PDL-1: Programmed cell death-Ligand 1; CXCR4: CXC chemokine receptor 4; IDO1: indoleamine 2,3-dioxygenase; CLTA-4: Cytotoxic T-lymphocyte-associated-protein 4; KIR2DL1/2/3: Killer cell Ig-like receptor; CSF1R: colony stimulating factor 1 receptor; CD40: Cluster of differentiation 40; TGFβ: Transforming growth factor beta; MEK: Mitogen activated Protein Kinase.

**Table 4 biomedicines-09-00389-t004:** Ongoing trials that are investigating the role of focal adhesion kinase (FAK) inhibitors in PDAC treatment.

FAK Inhibitor	Additional Therapy	Phase	NCT No.	Participants No.	Recruiting
Defactinib	Pembrolizumab+	I	NCT02546531	43	No
	Gemcitabine				
Defactinib	Pembrolizumab	I/II	NCT02758587	59	Yes
GSK2256098	Trametinib	II	NCT02428270	16	No

FAK: focal adhesion kinase.

## Data Availability

Not applicable.

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
