# Peer review of "The Latest Advancement in Pancreatic Ductal Adenocarcinoma Therapy: A Review Article for the Latest Guidelines and Novel Therapies"

_biomedicines, 2021, doi:10.3390/biomedicines9040389_

Round 1

Reviewer 1 Report

The authors have satisfactorily answered the questions of the reviewer. I believe that this is an interesting and valid manuscript.

Author Response

Thank you for all your support and recommendations, we very much appreciate it.

Reviewer 2 Report

I could not have access to a marked up version which rendered the revision of the MS more complicated.

In my opinion, authors answered to most questions. However, the following questions could be improved:

  • Justify the high focus on some RTK as it seems oversold as compared to other RTK pathways. What is the subclass of PDAC patients that could benefit from these treatments ?
  • 5 : explain why these clinical trials could have failed – Involvement of the fibrotic part of stroma is more complex than expected. As an example of this, the depletion of fibroblasts that produce the large amount of matrix accelerates pancreatic cancer development (PMID : 24856586, PMID : 24856585). Please discuss.
  • L322 : combination work better, but what are the general rules ?

Author Response

I could not have access to a marked up version which rendered the revision of the MS more complicated.

Thank you for all your support and recommendations, we very much appreciate it.

In my opinion, authors answered to most questions. However, the following questions could be improved:

  • Justify the high focus on some RTK as it seems oversold as compared to other RTK pathways. What is the subclass of PDAC patients that could benefit from these treatments?

tyrosine kinase inhibitors represent a newer generation of chemotherapeutic agents targeting specific tumor pathways associated with carcinogenesis (cell cycle control, signal transduction, apoptosis and angiogenesis). while Erlotinib is the prototype of the tyrosine kinase inhibitors with proven low efficacy in advanced pancreatic cancer, multiple other tyrosine kinase inhibitors targeting the VEGFR, PDGFR, and Src kinases are in various phases of clinical trials testing. Even though  for some trials the preliminary results have been not very promising, however, these trial provide future insights and guidance for improving patient selection, identifying effective combinations, improving the predictive value of current preclinical models and better study designs. 

  • 5: explain why these clinical trials could have failed – Involvement of the fibrotic part of stroma is more complex than expected. As an example of this, the depletion of fibroblasts that produce the large amount of matrix accelerates pancreatic cancer development (PMID: 24856586, PMID: 24856585). Please discuss.

Explanation was added and highlited in the updated manuscript’s version

  • L322: combination work better, but what are the general rules?

Unfortunately, pancreatic cancer cells become resistant to monotherapies by exclusion of drugs from cancer cells, by changes metabolizing enzymes; or by becoming more resistant to stress and apoptosis. Increasing levels of the drugs is limited by their somatic toxicity so numerous alternative therapies have been proposed. Testing these alternatives in clinical trials will be difficult unless they work with the standard treatments (like gemcitabine). most clinical trials have concentrated on combining different S-phase targeting agents with other agents. So, further incremental increase in survival benefit should be possible by targeting resistance to apoptosis, targeting stroma or even targeting multiple pathways in combination with gemcitabine.

This manuscript is a resubmission of an earlier submission. The following is a list of the peer review reports and author responses from that submission.

Round 1

Reviewer 1 Report

The article by Marwa Elsayed and Maen Abdelrahim reviews current treatment strategies for pancreatic cancer. The topic is timely and of interest, even though there are several current reviews about pancreatic cancer therapy. The manuscript is generally well written and up-to-date. There are some concerns the authors might want to address.

“accumulating genetic mutations in both somatic and germline cells”. What do the authors mean by germline cells in this context? There is no ‘accumulation’ in germline cells in hereditary cases.

“CONKO-001 was a pivotal randomized multicenter clinical trial”. While this trial was important, the ESPAC-1 trial established adjuvant chemotherapy as the standard therapy.

The phase III APACT trial (NCT01964430) should be included.

Some typos should be corrected, e.g. Resecrion, Kirestin-ras, 3ry

Author Response

“accumulating genetic mutations in both somatic and germline cells.” What do the authors mean by germline cells in this context? There is no ‘accumulation’ in germline cells in hereditary cases. The sentence was corrected accordingly.

“CONKO-001 was a pivotal randomized multicenter clinical trial”. While this trial was important, the ESPAC-1 trial established adjuvant chemotherapy as the standard therapy.  The word pivotal was changed. ESPAC-1 trial was mentioned already in L130.

The phase III APACT trial (NCT01964430) should be included. The trial is added.

Resecrion, Kirestin-ras typos are corrected. I didn't find 3ry typos

Reviewer 2 Report

In this review, the authors detail the current guidelines and open avenues on the novel therapeutic options for PDAC patients. The first part of the manuscript (guidelines) is very well written and represent very well the current status of the litterature. The second part of the manuscript (novel therapies) is less exhaustive and less balanced in the choice of litterature, and could be improved. As such, it could close the road to potential clinical developments or to potential basic and preclinical work of importance, in a manner that is not sufficiently justified in my opinion.

Major :

  • The authors should explain the method used to review the litterature and clinical trials, and position their work with regards to other litterature reviews in the same topic (update ? other angle ?).
  • L42 : PDAC physiopathology is still not understood.
  • L44-45 – does this sentence refer to all cancers ? the sentence is unclear.
  • L71 describe the theory of « non-evident micrometastasis » - is there basic, translational work on the topic ? are there specific possible therapies to tackle this issue ?
  • L93 – is there a molecular mechanism for this observation ?
  • L136 NAT – what are the major leads that are followed to possibly solve this controversy ?
  • L267 – in PDAC, are there clinical strategies (or pre-clinical) to specifically target each site of metastatic dissemination including peritoneal carcinomatosis ?
  • L328 : ref 72 is too general, focus on review on pancreatic cancer.
  • 1.1 : this whole paragraph is uncomplete, as it completely lacks the importance of PI3K signalling in PDAC. Basic research shows that the only KO or KD that is able to completely prevent pancreatic cancerogenesis is the inactivation of PI3Kalpha (PMID : 25452276 ; PMID : 25311989) ; other KO of EGFR, MEK, etc, did not present similar phenotypes. There is also a large preclinical litterature on therapies targeting downstream Kras signals, that is undercovered. Discuss the selection of patients ; in other cancers, when signal targeted therapies are used in a selected population of patients, the efficiency is clear. We are so far unable to do that for PDAC patients, so more work is needed in this area too, before ruling it out.
  • L336 : what is the percentage of PDAC with KrasG12C mutant ?
  • Justify the high focus on some RTK as it seems oversold as compared to other RTK pathways. What is the subclass of PDAC patients that could benefit from these treatments ?
  • paragraph 3.5 : explain why these clinical trials could have failed – Involvement of the fibrotic part of stroma is more complex than expected. As an example of this, the depletion of fibroblasts that produce the large amount of matrix accelerates pancreatic cancer development (PMID : 24856586, PMID : 24856585). Please discuss.
  • L322 : combination work better, but what are the general rules ?
  • Conclusion : genetic mutations will not suffice, as epigenetics is critical -discuss.

Minor : spelling issues & grammatical small issues

  • L 17 : Micrometastasis
  • L 22 : I do not understand the grammatical construction of the sentence « therefore … »
  • L242 : NLR LNR – it is unclear – explain why this ratio is important.
  • L276 : FOLFIRINOX
  • L292 : why a majuscule for « Twenty.. » ?
  • L312 : ration

Author Response

The authors should explain the method used to review the literature and clinical trials and position their work with regard to other literature reviews on the same topic. The method was added to the end of the abstract.

L42: PDAC physiopathology is still not understood. The concern was addressed and corrected in L45.

L44-45 – does this sentence refer to all cancers? the sentence is unclear. 

The concern was addressed and corrected in L47-48

L71: describe the theory of « non-evident micrometastasis » - is there basic, translational work on the topic? are there specific possible therapies to tackle this issue?  Both references number 22 and 23 mentioned that PDAC is a systemic disease that might have early metastatic foci that originate during the initial stage of PDAC formation and went undetectable at the time of diagnosis. This observation was based on a pre-clinical study performed by using a transgenic mouse model. Please refer to the following article for your reference "Rhim, A. D. et al. EMT and dissemination precede pancreatic tumor formation. Cell 148, 349–361 (2012)". NAT for resectable PDAC is an area of interest that is been currently under investigation, however, the current recommendation is to initiate adjuvant chemotherapy as early as possible after PDAC resection in order to eliminate any residual undetectable malignant foci. 

L93 – is there a molecular mechanism for this observation? There is no molecular mechanism behind this observation. This observation was concluded from the ESPAC-3 trial, a multicenter international randomized-controlled prospective phase III study. Data from another study (please refer to the following reference "Haeno H, Gonen M, Davis MB, et al.: Computational modeling of pancreatic cancer reveals kinetics of metastasis suggesting optimum treatment strategies. Cell 148:362-375, 2012") had emphasized that start timing of adjuvant chemotherapy following PDAC resection plays an important role in determining the median OS and PFS rates; however, it is important to notice that this study was designed in a retrospective fashion with the possibility of having biased data based on patient selection and administered adjuvant chemotherapy.

L136 NAT – what are the major leads that are followed to possibly solve this controversy? Prospective studies with a head-to-head comparison for both upfront surgical resection, and NAT followed by surgical resection for resectable PDAC are the best solution to address this controversy. Some of the ongoing trials are mentioned already in the article

L267 – in PDAC, are there clinical strategies (or pre-clinical) to specifically target each site of metastatic dissemination including peritoneal carcinomatosis? No such clinical data that is supporting this theory. However, the data from this study (please refer to reference No.59) support the superiority of FFX therapy compared to GnP for metastatic PDAC in regards to the median OS, even after adjusting the statistical analysis to the mentioned factors, e.g., number of metastatic sites, liver metastasis, and peritoneal carcinomatosis

L328: ref 72 is too general, focus on review on pancreatic cancer. The reference was changed (73)

1.1 : this whole paragraph is uncomplete, as it completely lacks the importance of PI3K signalling in PDAC. Basic research shows that the only KO or KD that is able to completely prevent pancreatic cancerogenesis is the inactivation of PI3Kalpha (PMID : 25452276 ; PMID : 25311989) ; other KO of EGFR, MEK, etc, did not present similar phenotypes. There is also a large preclinical litterature on therapies targeting downstream Kras signals, that is undercovered. Discuss the selection of patients ; in other cancers, when signal targeted therapies are used in a selected population of patients, the efficiency is clear. We are so far unable to do that for PDAC patients, so more work is needed in this area too, before ruling it out. This issue was addressed with more talk about PI3K, and RAF-MEK-ERK-MAPK. So far there are no definite criteria for patient selection before starting signal targeted therapies.  

L336 : what is the percentage of PDAC with KrasG12C mutant ? The % was added.

Justify the high focus on some RTK as it seems oversold as compared to other RTK pathways. What is the subclass of PDAC patients that could benefit from these treatments ? It was mentioned for the completion of the work

paragraph 3.5 : explain why these clinical trials could have failed – Involvement of the fibrotic part of stroma is more complex than expected. As an example of this, the depletion of fibroblasts that produce the large amount of matrix accelerates pancreatic cancer development (PMID : 24856586, PMID : 24856585). Please discuss. Discussed

L322 : combination work better, but what are the general rules ? There is no role, these were an observation from the clinical trials

Conclusion : genetic mutations will not suffice, as epigenetics is critical -discuss. Concern was addressed in L563

Minor : spelling issues & grammatical small issues

L 17 : Micrometastasis. Corrected

L 22 : I do not understand the grammatical construction of the sentence « therefore … » Corrected

L242 : NLR LNR – it is unclear – explain why this ratio is important. The ratio is not clinically important, thus, it was removed

L276 : FOLFIRINOX. Corrected

L292 : why a majuscule for « Twenty.. » ? Corrected

L312 : ration. Corrected
